# How Pendular Is Human Brachiation? When Form Does Not Follow Function

**DOI:** 10.3390/ani13091438

**Published:** 2023-04-22

**Authors:** Melody W. Young, James Q. Virga, Stratos J. Kantounis, Samantha K. Lynch, Noah D. Chernik, Jon A. Gustafson, Matthew J. Cannata, Nicholas D. Flaim, Michael C. Granatosky

**Affiliations:** 1Department of Anatomy, New York Institute of Technology College of Osteopathic Medicine, Old Westbury, New York, NY 11568, USA; 2Center for Biomedical Innovation, New York Institute of Technology College of Osteopathic Medicine, Old Westbury, New York, NY 11568, USA

**Keywords:** suspensory, locomotion, mechanical energy, center of mass, accelerometry

## Abstract

**Simple Summary:**

Brachiation is a form of suspensory (i.e., meaning the animal hangs below branches) locomotion in which only the forelimbs are used for weight support and propulsion. Brachiation has only ever evolved in Primates, and the living hominoids (i.e., apes) are considered specialized brachiating species. Odd among the apes are humans who have all but abandoned brachiation in favor of striding bipedalism. However, humans still retain the ability to adopt brachiation as a form of locomotion. As such, we explored the center of mass mechanics of human brachiation to explore how similar these movements matched the expectations of a simple pendulum. These findings were compared to data collected from non-human primates. Generally, humans demonstrated shorter than expected pendulum periods and remarkably low energy recovery compared to specialized brachiating species. We demonstrate that relatively long forelimb length and high grip forces, a proxy for global forelimb force-generating potential, act as the main driving factors to reduce energetic mechanical costs through effective pendular recovery.

**Abstract:**

Brachiation is a form of suspensory locomotion observed only in Primates. The non-human hominoids (e.g., gibbons, orangutans, chimpanzees, and gorillas) are considered specialized brachiators, yet peculiar among the living apes are anatomically modern humans (*Homo sapiens*), who have forgone this locomotor mode in favor of bipedal striding. Humans can, however, brachiate and seem to have retained the locomotor capabilities of their arboreal ancestors. However, the mechanics of human brachiation have not been quantified. In this study, we evaluate how closely human brachiation conforms to the expectations of simple pendular motion using triaxial accelerometry and high-speed videography. These data are compared to specialized brachiating non-human primates. We found that humans have lower energy recovery than siamangs (*Symphalangus syndactylus*) during brachiation and have shorter observed pendular periods than expected compared to other primates. We demonstrate that relatively long forelimb length and high grip forces, a proxy for global forelimb force-generating potential, act as the main driving factors to reduce energetic costs through effective pendular recovery. These data are the first to assess the strategies humans adopt to perform a behavior they are not anatomically specialized to execute and places them within a comparative framework amongst other brachiating primates. We show that although humans demonstrate behavioral flexibility during brachiation (e.g., differing mediolateral and vertical center of mass positional movement patterns), anatomical features are the primary driver of variation in brachiation performance.

## 1. Introduction

For most arboreal species, some form of suspensory locomotion (i.e., movement below branches) is inevitable when substrate size is small compared to the overall body dimensions [1,2,3]. Brachiation is a form of suspensory locomotion in which weight support and propulsion are achieved solely by the forelimbs [1,4,5]. Such a specialized locomotor mode has only evolved in Primates [5,6], and the propensity for brachiation is observed in nearly all living primate taxa (Figure 1; [4]). The most specialized living brachiators are the non-human hominoids (e.g., gibbons, orangutans, chimpanzees, and gorillas) [6,7,8]. As a group, these species are characterized by a range of anatomical specializations (e.g., long forelimbs, stiff axial skeletons, high mobility at the shoulder joint, and relatively large forelimb flexor musculature) often associated with improving locomotor performance while swinging below branches [7,9,10,11,12,13,14,15]. Odd among the hominoids are anatomically modern humans, which beyond habitual exercise and play [16], have all but abandoned brachiation as a form of locomotion in favor of specialized striding bipedalism. Such a disconnect between the ancestral [2,17,18,19] and modern conditions provides an interesting means to simultaneously test how neuromuscular locomotor patterns are conserved across phylogeny and whether a lack of anatomical specialization influences locomotor performance.

The fluid swinging motion observed when brachiators move beneath branches is similar to the oscillations of a pendulum and the repeated interchange of potential and kinetic energy [4,10,21,22,23,24]. In a perfect pendular system devoid of air resistance or mechanical friction, the magnitudes of kinetic and potential energy are equal yet opposite in magnitude so that their absolute maximum values coincide at the exact time [25,26]. During the swing, the pendulum effectively transforms the high initial potential into kinetic energy so that as potential energy reaches its minimum (e.g., at the lower arc of the swing), kinetic energy reaches its maximum [26]. In doing so, natural pendulums maximize energetic recovery through a seamless exchange between potential into kinetic energy, requiring no additional input of energy into the system other than the initial input of potential energy (i.e., lifting a pendulum to a starting position at an angle to the plane of gravity).

It is tempting, then, to assume brachiators mimic natural pendular motions, maximizing energy recovery (i.e., the amount of potential energy converted into kinetic energy to promote forward movement and reduce overall mechanical energetic costs) to reduce the muscular investment necessary to travel [27]. Limited studies have explored how closely brachiating primates do indeed match the expectations of a simple pendular model. Fleagle’s [28] seminal work on brachiation in siamangs (*Symphalangus syndactylus*) established the possibility of near-zero net work locomotion. However, such a conclusion was drawn from kinematics rather than accelerometry or force plates and only analyzed within a single stride. Additional studies [4,14,23,24,27,29] have demonstrated that brachiation can broadly be divided into two distinct speed-related center of mass (COM) patterns. At slower speeds, animals use gaits that are characterized by a doubled-limb support phase with duty factors of each limb >0.5, meaning at least one limb is always in contact with the support [14,27]. Arm-swinging gaits with continuous contact resemble, at some level simple pendular motion, where the animal’s COM moves along a sequence of circular arcs. However, the transition between arcs can be broadly classified based on four patterns of COM paths (i.e., point, loop, backward pendulum, and parabolic). A point transition is considered optimal, but all have been observed in brachiating gibbons [30]. At higher speeds, animals tend to use ricochetal gaits that are characterized by an aerial (non-contact) phase between support phases of each step [4,29]. Center-of-mass movements during ricochetal arm-swinging closely match a sine wave, wherein the trough coincides with the lowest position of the COM during the support phase [i.e., usually at mid-support (50% of support phase)], and the crest represents the highest point of the COM during a no-contact flight phase. Mechanical energy recovery during brachiation in gibbons is relatively high and ranges between 40–80% [4,24]. This range of energy recovery in gibbons is negatively correlated with speed but does not differ dramatically between continuous contact or ricochetal brachiation [24] (Figure 2b).

A well-established metric regarding pendulums is its period (T) or the amount of time necessary for the pendulum to complete an entire oscillation. This metric is affected by the length of the pendulum (L) and gravity alone. The pendular period is calculated as follows:$$T=2\pi \sqrt{\frac{L}{g}}$$
where T is the pendular period, and g is the gravitational acceleration (9.81 m/s^2^) [26]. This means lifting a pendulum 30° above the axis of gravity and lifting a pendulum 1° above the axis of gravity results in the same pendular period. This unintuitive yet highly consistent metric makes for an interesting comparison between expected and actual pendular periods in brachiators and allows us to further investigate exactly how closely brachiators are mimicking pendular mechanics. For the species analyzed thus far (i.e., douc langurs, gibbons, and spider monkeys), there is a tendency to demonstrate longer than expected pendulum periods regardless of experimental conditions [1,31].

In this study, we assess how closely the brachiation dynamics of anatomically modern humans match those predicted by simple pendular motion. Specifically, if human brachiation achieves pendular movement, we anticipate a near-perfect out-of-phase interchange of potential and kinetic energy resulting in high levels of energy recovery [4,23,24]. Further, we expect the human swing period duration to be determined by the vertical distance between the individual’s COM and the point of contact with the substrate [1,10], in other words, their L. These findings will be contextualized in a comparative framework using similar data collected from non-human primates [1,4,10,24,31]. Lastly, we assess the possible factors (i.e., forelimb length, body mass, grip force, a proxy for global forelimb force-generating potential [32,33,34,35], and mediolateral and vertical COM oscillations) contributing to brachiation costs in humans and rank the importance of these variables against each other. Such a study provides an elegant assessment of how a species modifies its behavior in circumstances where anatomical form does not match the intended functional goal. Although brachiation is not natural for humans, in these experiments, we can control aspects of subject performance and collect a range of anthropometric variables that are not usually possible for non-human primates. Such data can not only help us unravel the mechanics of brachiation in humans, but also aid our understanding of the anatomical correlates of brachiation broadly. Furthermore, there is limited information about brachiation dynamics across primates, and this study aims to contextualize this locomotor mode in a phylogenetic comparative context.

## 2. Methods

### 2.1. Subjects and Experimental Procedure

Human brachiation data were collected from nine participants (aged 23–40; average mass = 82.87 ± 14.24; male-to-female ratio = 7:2). Participants reported no pathologies related to pectoral, shoulder, or forelimb strength and movement. No previous climbing experience was reported among any participant. All protocols were approved by the New York Institute of Technology College of Osteopathic Medicine Internal Review Board (Protocol: BHS-1731). Data were collected under a single experimental condition using a two-meter-long set of horizontal monkey bars with an inter-bar distance of 0.135 m.

A triaxial accelerometer with a built-in gyroscope (Witmotion, Shenzhen, China) was attached to the dorsal aspect of the participant’s waistband, the approximate COM position in humans. Participants were instructed to traverse the monkey bars using one arm per rung and were allowed ample rest between trials. No other constraints/modifications were made to each participant’s brachiation style. Each participant contributed at least five runs. All trials were recorded with GoPro Hero10 (GoPro, San Mateo, CA, USA) from a lateral and frontal aspect at a frequency of 120 Hz. Accelerometer data and video footage were synchronized post hoc using large impulses generated by tapping the accelerometer prior to the start of each trial. Only strides in which the participant moved at an observable steady-state velocity were used in the final processing. Forelimb grip force (a proxy for global forelimb strength [32,33,34,35]) was measured using a dynamometer (South El Monte, CA). We followed the protocols established by [36], asking participants to sit upright with their arms at a ninety-degree angle with their palms facing inward. Participants were asked to squeeze at maximum strength on their right hand and alternate between their left and right hands between trials to prevent fatigue. Each participant contributed a total of three maximum grip strengths per hand, and these were averaged and adjusted for the body weight of each individual. Anthropogenetic measurements [i.e., height (m), weight (kg), and forelimb length (m)—measured from the tip of their middle finger to shoulder, and distance between the point of contact (fingertips) and position of the COM (L; see below)] were collected from each individual.

### 2.2. Data Processing

Spatiotemporal variables, including stride length (m), frequency (Hz) and time (s), velocity (m/s), and duty factor (%), were measured from lateral video footage of locomotor trials using ImageJ [37] from GoPro footage. Touchdown, liftoff, and subsequent touchdown frames of each stride were recorded and divided by the recording frame rate to obtain time. A stride was defined as a touchdown to the subsequent touchdown of a particular reference limb. Contact time was calculated by taking the difference between the initial touchdown and liftoff. The duty factor was calculated by dividing contact time by total stride time. Stride length was calculated by multiplying speed by stride time. Stride frequency is the inverse of stride duration. Speed was calculated by choosing a point on the participant’s head for each stride and determining the amount of time required to traverse a known distance on the runway. The expected pendular period was calculated as described above (see introduction).

Center of mass position and energy fluctuations were calculated following previously published methodology [38,39,40,41,42,43,44]. All accelerometer data were passed through custom MATLAB (Mathworks, Natick, MA, USA) code, taking the local system accelerations and associated Euler angles to mathematically derive accelerations in the global coordinate system post-detrending [45,46]. Accelerations were then integrated to first calculate instantaneous velocity and were secondarily integrated to identify the instantaneous position of the COM in each cardinal plane. To obtain constants for integration of velocity (initial fore-aft velocity), we digitized a consistent point on the most superior aspect of the participant’s head over stride for ten frames prior to the initial touchdown (see [43]). The mediolateral and vertical excursions were calculated as the difference between the minimum and maximum vertical or mediolateral positional values within a stride. Kinetic energy (EK) in the fore-aft (EK_x_), normal (EK_z_), and mediolateral (EK_y_) planes were calculated as follows:$$\mathrm{EK}=\frac{1}{2}mv^{2}$$
where *m* is the mass of the animal and *v* is the velocity of the COM in each respective plane. The potential energy of the COM (EP) was calculated using the following:EP=mgh
where *m* is the mass of the animal and *g* is gravitational force (9.81 m/s^2^), and *h* is the vertical displacement of the COM in the normal axis. Total fore-aft, mediolateral, and normal kinetic (EK) and potential energies (EP) were summed during each stride to obtain the total energy (ET) of the COM.

Pendular effectiveness was estimated as percent recovery following [47]:$$\mathrm{Percent}\;\mathrm{Recovery}\;\left(\%\right)=\frac{\left(\;\Delta \mathrm{EK}+\Delta \mathrm{EP}\right)-\Delta \mathrm{ET}}{\left(\;\Delta \mathrm{EK}+\Delta \mathrm{EP}\right)}\times 100$$
where ΔEK, ΔEP, and ΔET are the sum of the positive increments of the EK (+), EP (+), and ET (+) profiles, respectively.

All analyses were conducted in R (R Core Team 2013) using the packages ‘lmerTest’ [48], ‘lme4’ [49], and ‘qpcr’ [50,51]. The normality of datasets was assessed with Shapiro-Wilk and Levene’s tests [52] before rank-transforming all data to satisfy assumptions of subsequent statistical analyses [52]. Linear mixed-effect models were used to assess the effect of velocity, height-adjusted forelimb length, weight, body weight-adjusted grip force, and mediolateral and vertical excursions on energy recovery during brachiation. Additionally, individual behavioral idiosyncrasies were accounted for by including individuals as a random effect following [49,53] in all models. Iterative models containing every combination of fixed effects were run and compared to the model with the lowest Akaike’s information criteria (AIC). The change in AIC (ΔAIC) was calculated as the difference between the model run and the best-fit model. All ΔAICs under three were kept for further analysis. The subsequent AIC weights were determined and ranked the overall importance of each fixed effect on energy recovery [50].

## 3. Results

Stride parameters and mechanical energy variables are reported across individuals and are reported in Table 1. Forelimb length was, on average, 0.74 ± 0.05 m, and relative to height was 0.42 ± 0.01. Grip strength across individuals was 46.37 ± 8.68 kg, and relative to weight was 0.59 ± 0.13. During brachiation, humans move within a narrow speed range (0.60 ± 0.11 m/s) with large duty factors (71.16 ± 7.30%), stride frequencies of 0.84 ± 0.16 Hz, stride lengths of 0.69 ± 0.12 m, and greater mediolateral (0.33 ± 0.18 m) compared to vertical excursions (0.21 ± 0.14 m). No ricochetal brachiation (i.e., brachiation with an aerial phase) was detected across participants. The observed pendulum period (1.23 ± 0.25 s) during human brachiation was shorter than the expected period (2.21 ± 0.09 s; *p* < 0.001, Figure 2a). Human brachiation energy recovery was reported at 29.80 ± 11.26% and was not driven by speed differences (Figure 2b–d; *p* = 0.730, Table 1 and Table 2). We found the best predictors for high energy recovery during brachiation were relatively longer forelimb length and greater grip strength (a proxy for global forelimb force-generating potential [32,33,34,35]; Table 2, Figure 3a,b; both *p*-values < 0.002). Velocity, weight, and mediolateral and vertical excursions were not substantial predictors for high energy recovery (all *p*-values > 0.138, Table 2). In comparing the linear mixed effect models with Δ AIC values under three, we found that forelimb length and grip force were included in each of the eight models and held equal relative importance (importance = 1.0, Table 3). Four of the models included vertical excursion (Importance = 0.49), and two included mediolateral excursions, fore-aft velocity, and body mass (Importance = 0.19, 0.18, and 0.17, respectively; Table 3).

## 4. Discussion

It is not lost on the authors that brachiation in anatomically modern humans is likely of little fitness benefit. However, owing to a past evolutionary history where brachiation was at least present (Figure 1) based on specific anatomical features (e.g., grasping hands and mobile forelimbs [7,9,10,11,12,13,14,15]), humans retain the neuromuscular capability to adopt brachiation as a locomotor mode. The data presented in this study is one of the few to explore brachiation dynamics in a living primate species and, to our knowledge, is the only work directly correlating anatomical features to brachiation performance. As such, we test long-held form-function hypotheses about what makes an anatomically “ideal” brachiator. These data can be used in future studies to help explain why natural selection drives certain behavioral and anatomical specializations in habitual brachiating primates.

A study of this nature is, of course, faced with some limitations that must be considered. First, as human brachiation is not a natural behavior, it is possible that the novelty of the task forced individuals to adopt unnatural solutions to moving in this manner. However, as the goal of this study was to explore how humans effectively achieve brachiation, we do not consider the novelty of the task to be a limitation. It should be noted that all individuals had two practice sessions on the experimental runway before data collection, and each practice session was at least two weeks apart. Next, the experimental runway was relatively short. This experimental condition means that individuals could only achieve one or two strides. As such, it is unclear whether individuals would achieve greater energy recovery if they had the opportunity to increase the number of strides across the runway. We are unconvinced by such a possibility considering the relatively low range of energy recovery observed within the sample (Figure 2d). Further, the start-and-stop nature of the COM velocity (see below) indicates that each stride could be considered a nearly “unique” event implying little influence of increasing the length of the runway to accommodate additional strides. Related to the runway, the fixed length between adjacent bars means individuals were limited in varying their stride length. Such a limitation may have great influence, considering that forelimb length/total height was a major factor contributing to the energy recovery. Lastly, we chose to compare human brachiation to that of a simple pendulum for a few reasons. Biological correlates have historically been drawn between the act of brachiation and the simple pendulum, and this study intended to test these long-standing hypotheses in a comparative context [10,14,28,54]. Additionally, since the center of mass of the subject remained directly below the shoulder (which was used as the end of the pendulum), the restoring force acting on the subject is identical in magnitude and line of action as would be for a hypothetical point mass (i.e., the restoring force would point straight down and intersect the point mass at the end of the pendulum). That being said, we acknowledge there are inertial effects unaccounted for in modeling the period of human brachiation as a simple pendulum.

These limitations notwithstanding, the human subjects in our sample had relatively poor energy recovery compared to siamangs during brachiation (Figure 2d; [1,19]) and to striding bipedalism [55]. Human brachiation is highly variable between individuals in terms of spatiotemporal gait variables (Table 1) and overall COM mechanics (Figure 3a,b,d). Upon cursory visual inspection (Figure 2c), the potential and kinetic energy profiles between siamang and human brachiation appear morphologically similar, yet the sharp dip in kinetic energy at mid-stride during brachiation in humans implies a reduction in velocity in the middle of the stride and ultimately adds to the overall cost of locomotion. Contrary to the siamang trace, in which potential energy is effectively converted to kinetic energy (i.e., as potential energy reaches its minimum value, kinetic energy reaches its maximum value and assumes the primary role of driving the animal in the forward direction; Figure 2c), the exchange of potential into kinetic energy for humans is interrupted by a dip seen in the middle of the human energy trace (Figure 2c,d). Although not explicitly tested, it may be inferred that as humans are beginning their swing and gaining momentum, they fail to do so with the assistance of the already gained potential energy. Rather their bodies come to a near-stop and, therefore, minimum kinetic energy coincides with the time of minimum potential energy, clearly illustrating the disconnect in the potential to kinetic exchange. Therefore, humans are likely forced to input additional muscular effort to regain the lost momentum at mid-stride to continue propelling the body forward throughout the stride (Figure 2d). While this strategy is energetically costly from a mechanical perspective, limited pendular motion during suspensory movement is not unheard of in nature. Nyakatura and Andrada [56] demonstrated that two-toed sloths show no evidence of pendular movements during inverted quadrupedal walking. Such a strategy involves active deceleration of the COM during the downswing and, while costly from a mechanical and likely metabolic sense, allows for more controlled and careful locomotion.

We observed a range of variation in human brachiation regarding both tangential and mediolateral movements of the COM (Figure 3b,d). While some individuals adopt the pendulum-like falling and rising of the COM with relatively little mediolateral movements [4], others raised themselves upwards and/or demonstrated substantial mediolateral deviations (Figure 3b,d). While it is tempting to attribute energy recovery patterns to these gross fluctuations in COM movements, statistical models reveal neither tangential nor mediolateral excursions significantly influence variation in energy recovery (Table 2 and Table 3). This finding indicates that while humans adopt behavioral modifications to brachiate, such strategies have little influence on overall performance from a mechanical energy perspective.

The pendulum period in humans is shorter than expected based on *L*, a finding in opposition with observations in brachiating non-human primates (Figure 2a; [3]). Achieving simple pendular locomotion requires consistency in terrain (e.g., substrates spaced at equal distances, at the same height, with no obstruction, etc.), an environment simply nonexistent in nature [27]. It is unsurprising, then, that non-human primates opt for flexibility rather than efficiency to navigate the challenging arboreal conditions [27,54], resulting in pendular periods that do not match the expectations of a simple pendulum. Brachiation as a locomotor mode is well-known to result in relatively high rates of injuries compared to other modes of primate movement [57]. Slow and steady brachiation in primates allows more time for careful placement of the limbs and increases overall stability and security on thin arboreal substrates [58,59]. What remains unclear is why humans traversing a simple array of monkey bars show patterns of shorter-than-expected pendular periods. As energy recovery is low during human brachiation, this is likely an almost entirely muscularly driven behavior. It is highly plausible that as humans are traversing the terrain, they are applying torque at their shoulders and perhaps slightly swinging their bodies, which would easily drive movement faster than would be seen from the effects of gravity alone.

In this study, we created a natural experiment drawing from a range of subjects in terms of anatomical and performance differences to make direct comparisons between brachiating non-human primates. Specialized brachiating taxa, such as gibbons and spider monkeys, are well known to have elongated and highly muscular forelimbs [7,15,60,61]. Our most pendulum-like, as determined by energy recovery, human brachiators demonstrate a convergence [7,61] upon these morphological characteristics [i.e., relatively long forelimbs and greater grip force (a global proxy for forelimb musculature [32,33,34,35]) for their height/weight]. As discussed above, due to the sharp dip in kinetic energy at mid-stride, our subjects are likely forced to input additional muscular effort to regain the lost momentum to continue propelling the body forward. As such, individuals with increased muscle mass in the forelimb likely are better equipped to overcome the loss of momentum and maintain better energy recovery between strides. Of course, grip strength *per se* likely has little influence on the ability to provide forward momentum during human brachiation. Instead, such muscular propulsion would likely be provided by shoulder protractors/retractors [62]. Grip strength in this study is used as a global proxy [32,33,34,35] for forelimb (i.e., shoulder, arm, forearm, and intrinsic hand musculature) force-generating potential. This is an imperfect measure, and further testing utilizing electromyography paired with non-invasive metrics of muscle volume would help to test this hypothetical relationship. Further, it is important to note that while such a strategy may aid in achieving lower mechanical energy expenditure, it would have the opposite effect from a metabolic perspective (i.e., muscle activation is the primary driver of metabolic energy costs during locomotion [63,64]). The disconnect between mechanical and metabolic energy is a well-known phenomenon in studies of locomotion [40,65,66], and unfortunately, limited work has been conducted directly comparing the two directly. Likely human brachiation would be quite costly from a metabolic perspective [54], but such data has yet to be collected.

As for relative forelimb length, we speculate this correlation is likely due to the way mechanical energy recovery is calculated [4,24]. Values of energy recovery nearing 100% require the sum of the changes in potential and kinetic energy to largely exceed that of the total energetic change in the system, which should not fluctuate immensely unless there are large phase shifts between potential and kinetic energy. In a simple pendulum, changes in potential energy are essentially driven by the pendular arm length itself (e.g., longer pendular arms allow for greater height excursion and, in turn, larger changes in potential energy) [26]. At the bottom of the arc path, this potential energy is seamlessly transferred into kinetic energy resulting in a change in kinetic energy to be equal in magnitude to that of potential energy [26]. Longer relative forelimb lengths likely maximize the changes in potential energy, and while humans are not nearly as pendular as their hominoid relatives, there is still some exchange of potential into kinetic energy. Therefore, relative forelimb length likely drives energy recovery by maximizing the changes in potential and kinetic energy. Considering increased forelimb force-generating potential and relatively elongated forelimbs are characteristics of our best performing, as assessed by energy recovery, human brachiators, we suggest these data lend strong support that relatively long and strong forelimbs are critical adaptations for mechanically efficient brachiation. Such a finding explains the convergent morphology [7,9,10,11,12,13,14,15] that has arisen independently among distantly related brachiating primates.

## 5. Conclusions

From a gross motor perspective, human brachiation follows basic gait characteristics observed in specialized arm-swinging species, such as gibbons and spider monkeys. However, the overall pendular performance and energy recovery pale in comparison. During brachiation, humans have low energy recovery (~30%) and shorter than expected pendular periods. Increased energy recovery during brachiation across subjects is largely driven by anatomical features, such as relatively long forelimbs and high force-generating potential in the forelimbs. Despite variation in COM positioning during brachiation (Figure 2 and Figure 3), these have little influence on the overall performance. Such findings provide a natural experiment to help explain the similar morphology that has arisen independently between specialized brachiating primates and provide an elegant example of how neuromuscular flexibility can assist animals in executing a novel behavior when the form does not follow function.

## Figures and Tables

**Figure 1 animals-13-01438-f001:**
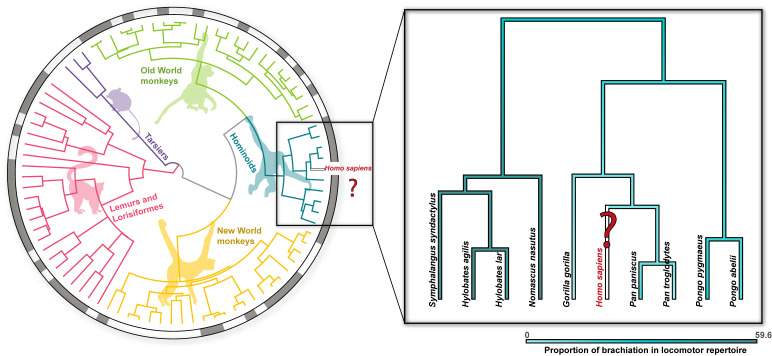
The presence of brachiation (gray squares) mapped on the primate phylogenetic tree. Of the living primates, brachiation is most common in hominoids (inset) and accounts for nearly 60% of the locomotor repertoire of some hylobatids. The proportion of brachiation in the locomotor repertoire of each species is denoted by variations in blue shading (see bottom right legend). Brachiation is notably absent as a natural locomotor behavior in anatomically modern humans. The phylogenetic tree was generated from TimeTree5 [20], and locomotor behavior and percentages were extracted from Granatosky [6].

**Figure 2 animals-13-01438-f002:**
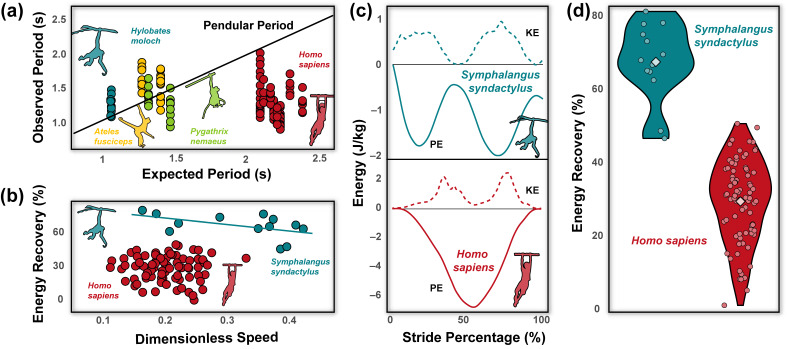
Across primate brachiators, the human pendulum period tends to be shorter than predicted values based on the length between the point of contact on the substrate and the center of mass (**a**). The black line represents a one-to-one relationship between the predicted and expected pendulum period. Data of pendulum period from non-human primates modified from Byron and colleagues [1]. Energy recovery is speed dependent in siamangs; however, speed does not drive energy recovery in human brachiators (**b**). The linear dimension used to calculate dimensionless speed was forelimb length following Michilsens and colleagues [24]. While potential and kinetic energy profiles between siamang (*Symphalangus syndactylus*) and human brachiation appear similar upon visual inspection (**c**), the sharp dip in kinetic energy during brachiation in humans implies individuals tend to reduce velocity in the middle of the stride. Illustration of potential and kinetic energy from siamangs recreated by Michilsens and colleagues [30]. Energetic traces from humans are representative traces (see Figure 3 for average traces). Such deceleration puts kinetic and potential energy peaks in phase with one another, resulting in relatively low energy recovery during human brachiation compared to siamangs (**d**). Circles represent individual data points, while colored diamonds reflect the mean of each sample. Energy recovery data from siamangs from Michilsens and colleagues [24].

**Figure 3 animals-13-01438-f003:**
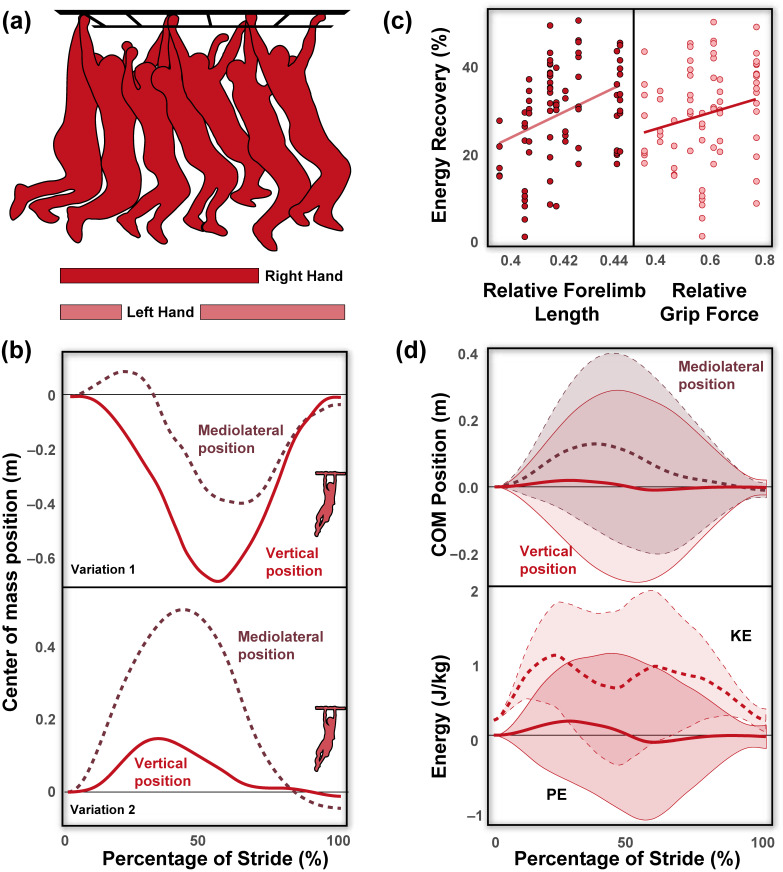
Human brachiation is a slow gait that involves continuous contact with the forelimbs, as seen in the gait cycle (**a**). Human brachiation is characterized by a range of strategies regarding both tangential and mediolateral movements of the center of mass seen in these representative traces (**b**). While some individuals adopt the pendulum-like falling and rising of the center of mass with relatively little mediolateral movements (Variation 1), others raise themselves upwards and/or demonstrate substantial mediolateral deviations (Variation 2). Despite these differences in center of mass movements during brachiation, the primary predictor of energy recovery (%) is driven by the forelimb-to-height ratio and relative grip strength, with individuals that have relatively longer and stronger forelimbs achieving greater energy recovery during brachiation (**c**). Average center of mass positional traces (in meters) with shaded standard deviation bars in the vertical and mediolateral planes (top) across a stride (**d**). Average kinetic (KE) and potential (PE) energy (in J/kg) traces with shaded standard deviation bars (bottom) across a stride. Standard deviation bars are enclosed with the same line marking as the average trace line (e.g., mediolateral position standard deviation is enclosed in dashed lines, matching the mean trace).

**Table 1 animals-13-01438-t001:** Anthropogenetic measurements [i.e., height (m), weight (kg), forelimb length (m), ratio of forelimb length to height, ratio of grip strength to body weight], spatiotemporal gait variables [i.e., average fore-aft velocity (m/s), duty factor, stance time (s), swing time (s), stride frequency (Hz), stride length (m)], energetic and pendular variables [i.e., energy recovery (%), expected and actual pendulum periods (s), mediolateral and vertical excursions (m)], represented as means ± standard deviations collected from *N* = 9 human subjects (*Homo sapiens*). Number of strides contributions is denoted by *n*.

Individual	1	2	3	4	5	6	7	8	9
*n*	19	10	5	10	6	5	7	8	10
Height [m]	1.7	1.75	1.68	1.85	1.83	1.75	1.75	1.93	1.79
Weight [kg]	65.77	85.73	66.22	79.38	102.15	74.39	80.2	100.24	97.52
Forelimb Length [m]	0.7	0.77	0.66	0.75	0.76	0.74	0.71	0.85	0.76
Forelimb Length/Height	0.41	0.44	0.39	0.4	0.42	0.42	0.41	0.44	0.43
Grip Force/Body Weight	0.77	0.53	0.47	0.57	-	0.42	0.64	0.36	0.61
Avg Foreaft Velocity [m/s]	0.59 ± 0.11	0.54 ± 0.04	0.49 ± 0.05	0.66 ± 0.06	0.44 ± 0.05	0.50 ± 0.04	0.68 ± 0.06	0.49 ± 0.08	0.70 ± 0.06
Duty Factor [%]	62.02 ± 4.61	71.95 ± 3.71	77.54 ± 2.59	73.87 ± 1.58	81.72 ± 2.19	79.72 ± 2.27	72.10 ± 2.5	76.36 ± 3.64	66.10 ± 3.13
Stance Time [s]	0.85 ± 0.17	0.84 ± 0.11	1.10 ± 0.14	0.78 ± 0.07	1.22 ± 0.14	1.10 ± 0.1	0.74 ± 0.09	0.97 ± 0.11	0.62 ± 0.06
Swing Time [s]	0.53 ± 0.16	0.32 ± 0.03	0.32 ± 0.03	0.27 ± 0.03	0.27 ± 0.02	0.28 ± 0.01	0.28 ± 0.01	0.30 ± 0.05	0.32 ± 0.03
Stride Frequency [Hz]	0.75 ± 0.15	0.87 ± 0.07	0.71 ± 0.07	0.96 ± 0.09	0.67 ± 0.06	0.73 ± 0.05	0.98 ± 0.09	0.79 ± 0.07	1.07 ± 0.07
Stride Length [m]	0.80 ± 0.19	0.62 ± 0.06	0.69 ± 0.00	0.69 ± 0.00	0.66 ± 0.05	0.69 ± 0.00	0.69 ± 0.00	0.62 ± 0.08	0.65 ± 0.03
Energy Recovery [%]	33.99 ± 10.42	36.06 ± 7.84	19.75 ± 5.45	15.63 ± 10.29	29.21 ± 10.74	28.39 ± 5.31	30.07 ± 6.55	28.08 ± 9.43	37.07 ± 11.08
Predicted Pendulum Period [s]	2.10	2.10	2.12	2.24	2.30	2.16	2.22	2.40	2.24
Pendulum Period [s]	1.39 ± 0.31	1.16 ± 0.10	1.41 ± 0.14	1.05 ± 0.10	1.49 ± 0.13	1.38 ± 0.10	1.03 ± 0.09	1.27 ± 0.12	0.94 ± 0.06
Mediolateral Excursion [m]	0.45 ± 0.17	0.25 ± 0.15	0.39 ± 0.17	0.40 ± 0.14	0.39 ± 0.18	0.37 ± 0.12	0.10 ± 0.08	0.40 ± 0.13	0.13 ± 0.05
Vertical Excursion [m]	0.24 ± 0.11	0.10 ± 0.06	0.31 ± 0.10	0.14 ± 0.06	0.38 ± 0.22	0.23 ± 0.15	0.17 ± 0.09	0.27 ± 0.12	0.13 ± 0.07

**Table 2 animals-13-01438-t002:** Statistical parameters derived from least squares regressions demonstrate the statistical importance of average fore-aft velocity, forelimb length (as a ratio of body height), grip force (as a ratio of body mass), mediolateral and vertical excursions, and body mass on energy recovery during human brachiation. Significant *p*-values are in bold.

Response Variable	Fixed Effect	Estimate	Standard Error	df	*t* Value	*p*-Value
Energy Recovery %	Average Fore-aft Velocity [m/s]	10.86	31.32	74.00	0.35	0.730
	Forelimb/Height Ratio	895.76	231.88	74.00	3.86	**<0.001**
	Weight [kg]	−0.03	0.34	74.00	−0.10	0.920
	Mediolateral Excursion [m]	−7.20	14.35	74.00	−0.50	0.617
	Vertical Excursion [m]	31.88	21.27	74.00	1.50	0.138
	Body weight adjusted grip force	82.29	24.96	74.00	3.30	**0.002**

**Table 3 animals-13-01438-t003:** Top eight models (with ΔAIC < 3) used in determining the variables that contribute the most to high energy recovery during human brachiation. Plus signs (+) indicate the variable was included in the model. All models included both height-adjusted forelimb length and body weight-adjusted grip force. Four included vertical excursions, and two included mediolateral excursions, fore-aft velocity, and body mass. AIC, Akaike’s information criterion.

ΔAIC	Forelimb Length	Grip Force	Vertical Excursion	Mediolateral Excursion	Fore-Aft Velocity	Body Mass
0	+	+				
0.2274	+	+	+			
1.7685	+	+	+	+		
1.8837	+	+		+		
1.8954	+	+	+		+	
1.9957	+	+				+
2	+	+			+	
2.1012	+	+	+			+
Importance	1.00	1.00	0.49	0.19	0.18	0.17

## Data Availability

All data used for statistical analysis within this study are presented as Appendix A.

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
