# Peer review of "How Pendular Is Human Brachiation? When Form Does Not Follow Function"

_animals, 2023, doi:10.3390/ani13091438_

Round 1
Reviewer 1 Report
Comments to Authors
In this article the authors study the determinants of pendular energy recovery in humans during brachiation. They measured center of mass motion and a number of anthropometric variables in nine human subjects brachiating on a set of monkey bars, and used statistical models to determine which variables best predicted energy recovery. They also compared their results for humans to those of other primates. This is a novel study that uses humans as a model to study a behavior that is unique to primates, and is able to test some possible anatomical determinants of brachiation performance. My biggest concern about this study is that the authors do not make the overall purpose of the study clear. I detail this and other concerns in my comments.
General Comments
I think the authors need to do more to explain the purpose of this research in their introduction. As they mention in the paper, brachiating is not a natural human form of locomotion, but we’re capable of it likely because we evolved from a brachiating ancestor. So this research doesn’t really hold any clear relevance for modern humans. Perhaps the authors could say more in the introduction about how these experiments might tell us more about the biomechanics of brachiating? Although brachiation isn’t natural for humans, in these experiments the authors have the ability to control aspects of subject performance, and also take a lot of anthropometrics that are hard to get in nonhuman primates. If the authors decide to take this approach, they need to be more explicit about what this analysis could potentially tell us about brachiation that previous studies on natural brachiators like siamangs have not. The authors do mention something about this study showing how species modify behavior when anatomical form is not adapted for the relevant function (lines 119-121 and in the abstract), but this idea is pretty vague. Maybe they could elaborate on this a little more too?
The authors end up investigating several anthropometric variables that are believed to be adaptations for brachiation in nonhuman primates, forearm muscle strength and arm length, but they do not make it clear why these variables are expected to improve mechanical energy recovery. In the case of forearm muscle strength, they suggest that this might help add energy back to the swinging body to maintain forward velocity and prevent a dip in kinetic energy mid-swing (lines 328-333). How would forearm muscles be capable of doing this? They are mainly used for grip, but I would think that shoulder musculature would be recruited to help swing the body forwarded, and I can’t see what the forearm muscles would do unless the authors think the wrist is somehow involved. Additionally, while this might result in a smoother kinetic energy curve, it would do so by requiring metabolic energy expenditure to activate muscles, and therefore it might not actually reduce energy costs of brachiation, because it wouldn’t be a passive exchange of kinetic and potential energy. This issue should be addressed too.
Additionally, the authors suggest that a longer arm might be beneficial for brachiation economy because it would increase stride length, thereby reducing the number of strides needed to travel a given distance (lines 333-337). This may be true, but I do not see how this is linked to pendular energy recovery. I think the authors need to try to explain what the connection between arm length and energy recovery more specifically is, since those are the two variables that are correlated here, whereas cost of transport was not investigated.
One last general concern I have is the extent to which the experimental setup may have affected the subjects’ performance. The monkey bar row is described as 2 meters long, and based on stride lengths this would have limited most subjects to only 2 complete strides per recorded bout. Might the results have been different if subjects could travel further? Perhaps better energy recovery could have been achieved if subjects were able to get into a better rhythm. I also wonder whether low energy recovery might have been related to the novelty of the task. Would subjects have achieved more efficient brachiation with more practice? I think the authors should discuss these as potential limitations of this study, and also maybe say more about subject experience with brachiation, as well as how many complete strides subjects were able to achieve per bout.
Specific comments
Line 156 – How was speed measured?
Lines 169-171 – Say more about initial velocity calculation. What part of the head was used, and what interval of the swing was used for the calculation. Also, Riskin citation is not listed in references.
Lines 195-196 – Provide numbers for these citations. Are they in the references?
Lines 278-280 – I believe this is a sentence fragment.
Line 300 – I think this should be Figure 2b, not 2a.
Lines 309-311 – Clarify how a longer pendular period is related to locomotor flexibility (rather than efficiency).
Table 1 – Change medialateral to mediolateral. Also, excursion distances are reported in units of degrees, but I think they’re supposed to be in units of meters.
Table 3 – Please describe results of AIC tests in the Results text.
Figure 1a – This is a cool figure, although it could be explained a little bit more. What is a ‘trait value,’ and how were the colors generated? Do the color shades represent reconstructions of % of locomotor time spent brachiating? Also, it would probably be helpful to refer to this figure earlier in the text.
Figure 1c – Are these average values across subjects, or just representative values from a single trial? Either way, it would be helpful to see variation from the data collected for this study plotted in this fashion. Either all strides could be plotted, or standard deviation bars could be added to these plots.
Figure 2b – What are the Y and Z axes depicted here? Can you replace ‘Y position’ and ‘Z position’ with ‘Mediolateral’ and ‘Vertical’ position to make this more clear? It would also be useful to see plots showing all of the variation among subjects (see comment above).
Author Response
Reviewer 1
Comments and Suggestions for Authors
Comments to Authors
In this article the authors study the determinants of pendular energy recovery in humans during brachiation. They measured center of mass motion and a number of anthropometric variables in nine human subjects brachiating on a set of monkey bars, and used statistical models to determine which variables best predicted energy recovery. They also compared their results for humans to those of other primates. This is a novel study that uses humans as a model to study a behavior that is unique to primates, and is able to test some possible anatomical determinants of brachiation performance. My biggest concern about this study is that the authors do not make the overall purpose of the study clear. I detail this and other concerns in my comments.
General Comments
I think the authors need to do more to explain the purpose of this research in their introduction. As they mention in the paper, brachiating is not a natural human form of locomotion, but we’re capable of it likely because we evolved from a brachiating ancestor. So this research doesn’t really hold any clear relevance for modern humans. Perhaps the authors could say more in the introduction about how these experiments might tell us more about the biomechanics of brachiating? Although brachiation isn’t natural for humans, in these experiments the authors have the ability to control aspects of subject performance, and also take a lot of anthropometrics that are hard to get in nonhuman primates. If the authors decide to take this approach, they need to be more explicit about what this analysis could potentially tell us about brachiation that previous studies on natural brachiators like siamangs have not. The authors do mention something about this study showing how species modify behavior when anatomical form is not adapted for the relevant function (lines 119-121 and in the abstract), but this idea is pretty vague. Maybe they could elaborate on this a little more too?
We thank the reviewer for this comment. Upon rereading the manuscript, we agree that our original version was a bit lax concerning what this study adds to the larger scientific literature. We hope the revised text in the introduction and discussion helps to make a stronger argument.
The authors end up investigating several anthropometric variables that are believed to be adaptations for brachiation in nonhuman primates, forearm muscle strength and arm length, but they do not make it clear why these variables are expected to improve mechanical energy recovery. In the case of forearm muscle strength, they suggest that this might help add energy back to the swinging body to maintain forward velocity and prevent a dip in kinetic energy mid-swing (lines 328-333). How would forearm muscles be capable of doing this? They are mainly used for grip, but I would think that shoulder musculature would be recruited to help swing the body forwarded, and I can’t see what the forearm muscles would do unless the authors think the wrist is somehow involved. Additionally, while this might result in a smoother kinetic energy curve, it would do so by requiring metabolic energy expenditure to activate muscles, and therefore it might not actually reduce energy costs of brachiation, because it wouldn’t be a passive exchange of kinetic and potential energy. This issue should be addressed too.
We completely agree with the reviewer and added text in the discussion addressing both points. We are not implying that grip strength per se is related to energy costs, but can be used as a proxy, albeit imperfect, for global forelimb strength. Further, we elaborated on the disconnect between mechanical and metabolic energy profiles.
Additionally, the authors suggest that a longer arm might be beneficial for brachiation economy because it would increase stride length, thereby reducing the number of strides needed to travel a given distance (lines 333-337). This may be true, but I do not see how this is linked to pendular energy recovery. I think the authors need to try to explain what the connection between arm length and energy recovery more specifically is, since those are the two variables that are correlated here, whereas cost of transport was not investigated.
Upon rereading our initial submission, we completely agree that our initial argument was insufficient. We have expanded the discussion to better address the relationship between relative arm length and energy recovery.
One last general concern I have is the extent to which the experimental setup may have affected the subjects’ performance. The monkey bar row is described as 2 meters long, and based on stride lengths this would have limited most subjects to only 2 complete strides per recorded bout. Might the results have been different if subjects could travel further? Perhaps better energy recovery could have been achieved if subjects were able to get into a better rhythm. I also wonder whether low energy recovery might have been related to the novelty of the task. Would subjects have achieved more efficient brachiation with more practice? I think the authors should discuss these as potential limitations of this study, and also maybe say more about subject experience with brachiation, as well as how many complete strides subjects were able to achieve per bout.
This is an excellent point. We have added a limitations section to address this and other potential issues with the study.
Specific comments
Line 156 – How was speed measured?
Speed was calculated by choosing a point on the participants head for each stride and determining the amount of time required to traverse a known distance on the runway.
Lines 169-171 – Say more about initial velocity calculation. What part of the head was used, and what interval of the swing was used for the calculation. Also, Riskin citation is not listed in references.
We have fixed added information about initial velocity calculation and fixed the missing Riskin citation.
Lines 195-196 – Provide numbers for these citations. Are they in the references?
We have fixed the references.
Lines 278-280 – I believe this is a sentence fragment.
Thank you for this catch. This has been corrected.
Line 300 – I think this should be Figure 2b, not 2a.
Thank you for this catch. This has been corrected.
Lines 309-311 – Clarify how a longer pendular period is related to locomotor flexibility (rather than efficiency).
We agree that flexibility does not mean slower speeds per se, but more generally not matching expectations of a simple pendulum. We have revised accordingly. We have also mentioned why slower speeds are likely to be better linked with increasing stability. We have revised.
Table 1 – Change medialateral to mediolateral. Also, excursion distances are reported in units of degrees, but I think they’re supposed to be in units of meters.
Thank you for this catch. These have been corrected.
Table 3 – Please describe results of AIC tests in the Results text.
This has been added to the main results text.
Figure 1a – This is a cool figure, although it could be explained a little bit more. What is a ‘trait value,’ and how were the colors generated? Do the color shades represent reconstructions of % of locomotor time spent brachiating? Also, it would probably be helpful to refer to this figure earlier in the text.
Thank you for this comment. Trait value in this instance is the proportion of brachiation in the locomotor repertoire. We have thus replaced ‘trait value’ with this to clarify what the colors mean. This figure was generated in R using the ‘phytools’ package. It is also referenced within the first paragraph of the introduction.
Figure 1c – Are these average values across subjects, or just representative values from a single trial? Either way, it would be helpful to see variation from the data collected for this study plotted in this fashion. Either all strides could be plotted, or standard deviation bars could be added to these plots.
Thank you for this comment. As we were trying to illustrate the difference between the Siamang and human energetic traces using the representative trace provided by Bertram and Chang, we followed suit in illustrating a representative trace from our own human data. We are electing to keep the figure in the main manuscript, however we have added an additional figure (Figure 3) with the average trace plotted along with shaded standard deviations. We hope this helps readers see the large range of variation existing in brachiation in humans.
Figure 2b – What are the Y and Z axes depicted here? Can you replace ‘Y position’ and ‘Z position’ with ‘Mediolateral’ and ‘Vertical’ position to make this more clear? It would also be useful to see plots showing all of the variation among subjects (see comment above).
Thank you for this comment. We have changed the Y/Z position with the mediolateral and Vertical position. We have also updated the Y axes label to be Center of mass position (m) and added “Variation 1” and “Variation 2” to clarify these are COM position traces in 2 varieties. We have also added an additional figure (Figure 3) with the average trace plotted along with shaded standard deviations. We hope this helps readers see the large range of variation existing in brachiation in humans.

Reviewer 2 Report
The context of why brachiation is important to study can be improved. Additionally, any information on any climbing/related experience of the subjects seems necessary when assessing forelimb-dominated brachiation biomechanics in a small group of humans. Figures and presentation were clear and helpful.
Author Response
Reviewer 2
Comments and Suggestions for Authors
The context of why brachiation is important to study can be improved. Additionally, any information on any climbing/related experience of the subjects seems necessary when assessing forelimb-dominated brachiation biomechanics in a small group of humans. Figures and presentation were clear and helpful.
We thank the reviewer for these comments. Accordingly, we have increased the attention in the introduction to explain why brachiation is important and we have provided information about climbing history.

Reviewer 3 Report
Dear authors,
I had the opportunity to revise your manuscript entitled « How pendular is human brachiation? When form does not follow function. »
I really enjoyed reading your study and the perspectives it opens. The manuscript is well written, and the topic is very interesting for the field of biological anthropology.
Before considering this manuscript publishable, I have one main concern about the analyses. I'm confused with the calculation of the “expected pendular period”. You write that “Pi rather than 2Pi is used to assess pendular period in the forward direction and not the remaining back swing as we were interested in forward contributing movement rather than the passive backswing observed in natural pendulum”, but a full stride should actually cover a full pendular period. There are two steps in a stride, i.e. two swing phases, so we should expect 2 times a pendular motion in the forward direction during a stride. That’s said, the stride is not clearly defined by the authors. Line 152, it is written that “touchdown, liftoff and subsequent touch-down frame of each stride were recorded…” so I believe that the stride should cover two times half a period, i.e., equivalent to a full pendular (back and forth) period by stride. This needs to be clarified because it could change the results with the predicted pendulum period being much longer than the pendulum period (if it’s calculated for the full stride). I would like the authors to consider this important methodological aspect. Otherwise, I have few additional comments.
Introduction:
l.85-86, You should add some description of the continuous sequence by at least including the idea that different forms of transitions have been described (i.e. point, loop, backward pendulum, parabolic). The following article should be cited: Michilsens F, D'Août K, Vereecke EE, Aerts P (2012) One step beyond: Different step-to-step transitions exist during continuous contact brachiation in siamangs. Biology Open 1(5):411-421
l.93-94, It is not very clear here because it looks like ricochetal brachiation is always related to higher speed and low energy recovery, but it’s not what Michilsen et al. (2011) show. During ricochetal brachiation the energy recovery is lower (7-8%) but it’s not significantly different than the energy recovery during brachiation with continuous contact. Furthermore, it is the speed that affect energy recovery (in siamang) regardless of brachiation style. I encourage you to provide these subtilities here.
l.97, I would remove the term “curious” as it gives an odd impression. This is a well checked and assessed metric.
l.101-103, It’s actually not fully exact, there is no influence of the initial angle if the angular displacement of the pendulum is kept small. But in the context of lifting a pendulum by 60° above axis of gravity, you’ll see a difference with the 1° lift. To keep this sentence more exact you should change 60° for 30° for example. Otherwise, provide more explanation about the mechanics of the pendulum.
I’m also wondering why the authors did not consider using the period of a physical pendulum instead of, or in addition to, a simple pendulum. I think this should be discussed because, at first glance, it is unlikely that the brachiation dynamics of human would follow a simple pendular motion anyway. Inertia is likely to play a certain role.
l.117, In the possible factors assessed, I’m wondering why you didn’t take into account the distance between the CoM and the point of contact. I guess that arm length can be considered as a proxy for this, but postural adjustments, such as flexing the legs for example, could have a direct influence on the position of the CoM. Did you abserve this kind of postural strategy?
Material and Methods
l.124, The average mass of the participants should be reported, as well as the sex ratio.
l.156, Stride frequency is the inverse of stride duration (not stride length).
l.160-162, To be clarified according to the definition of the stride.
l.164, I think it would be better to develop further on the calculation of the COM position.
l.176, I would keep the term “plane” instead of “axis”.
l.180, Change “9.81 m/s” by “9.81 m/s2”
l.185, The multiplication sign looks like the letter “x”
l.199, Why did you keep the models with the delta AICs under three? Is it arbitrary, or based on literature?
Results
l.203, The first sentence is odd. I think a "and" is missing.
l.206, The unit for the grip strength is missing.
l.209, The unit for the stride length is missing
l.211. Is the observed pendulum period calculated per stride, or per swing phase?
In Table 1, I would find interesting to add the absolute grip forces.
In Table 1, change “Medialateral” for “Mediolateral”
In Table 1, The unit of the mediolateral and vertical excursions are in °, but it should be in meter.
Figure 1.c) and Figure 2.b) It would be informative to include the “handfall pattern” on the graphs, at least vertical lines showing the timing of the lift-off for each hand and the double support.
Discussion
l.268, References are missing about the specific anatomical features related to brachiation in hominins.
l.276, You write “The sharp dip in kinetic energy at mid-swing”, while in the legend of Figure 1 you write “…individuals tend to reduce velocity in the middle of the stride”. So, this is really confusing because the mid-stride should be a phase of double support, while mid-swing should represent a period where there is only one support. It is important to define the stride and to refer to the stride (i.e. two swing periods) or to the step (i.e. one swing period), when appropriate.
l.299, Is such a strategy called “brachiation”? It looks more like a “flexed elbow forelimb swing” as it is defined in Hunt et al. 1996 (Hunt KD, Cant J, Gebo D, Rose M, Walker S, Youlatos D (1996) Standardized descriptions of primate locomotor and postural modes. Primates 37(4):363-387). To be discussed.
l.309, What about security in addition to flexibility?
l.327, Some references for this convergent morphology should be added here.
I think that a figure showing the relationship between dimensionless speed and energy recovery in humans and in gibbons and/or siamang would be highly valuable and informative in this study (even if it's not significant in humans). I believe that the authors have all the relevant data to do so.
Author Response
Reviewer 3
Comments and Suggestions for Authors
Dear authors,
I had the opportunity to revise your manuscript entitled « How pendular is human brachiation? When form does not follow function. »
I really enjoyed reading your study and the perspectives it opens. The manuscript is well written, and the topic is very interesting for the field of biological anthropology.
Before considering this manuscript publishable, I have one main concern about the analyses. I'm confused with the calculation of the “expected pendular period”. You write that “Pi rather than 2Pi is used to assess pendular period in the forward direction and not the remaining back swing as we were interested in forward contributing movement rather than the passive backswing observed in natural pendulum”, but a full stride should actually cover a full pendular period. There are two steps in a stride, i.e. two swing phases, so we should expect 2 times a pendular motion in the forward direction during a stride. That’s said, the stride is not clearly defined by the authors. Line 152, it is written that “touchdown, liftoff and subsequent touch-down frame of each stride were recorded…” so I believe that the stride should cover two times half a period, i.e., equivalent to a full pendular (back and forth) period by stride. This needs to be clarified because it could change the results with the predicted pendulum period being much longer than the pendulum period (if it’s calculated for the full stride). I would like the authors to consider this important methodological aspect. Otherwise, I have few additional comments.
Thank you for the reviewer’s kind words. We agree the analyses initially conducted was erroneous in the fact that it was only accounting for half the stride, rather than the entire touch down to touch down catching this. We have recalculated our expected pendulum periods using this 2pi instead of pi and updated all figures and analyses in the MS.
Introduction:
l.85-86, You should add some description of the continuous sequence by at least including the idea that different forms of transitions have been described (i.e. point, loop, backward pendulum, parabolic). The following article should be cited: Michilsens F, D'Août K, Vereecke EE, Aerts P (2012) One step beyond: Different step-to-step transitions exist during continuous contact brachiation in siamangs. Biology Open 1(5):411-421
Following the reviewer’s comment, we have added some text related to these different COM paths.
l.93-94, It is not very clear here because it looks like ricochetal brachiation is always related to higher speed and low energy recovery, but it’s not what Michilsen et al. (2011) show. During ricochetal brachiation the energy recovery is lower (7-8%) but it’s not significantly different than the energy recovery during brachiation with continuous contact. Furthermore, it is the speed that affect energy recovery (in siamang) regardless of brachiation style. I encourage you to provide these subtilities here.
We thank the reviewer for pointing out this caveat. We have corrected the text accordingly.
l.97, I would remove the term “curious” as it gives an odd impression. This is a well checked and assessed metric.
Thank you. We have removed it from the MS.
l.101-103, It’s actually not fully exact, there is no influence of the initial angle if the angular displacement of the pendulum is kept small. But in the context of lifting a pendulum by 60° above axis of gravity, you’ll see a difference with the 1° lift. To keep this sentence more exact you should change 60° for 30° for example. Otherwise, provide more explanation about the mechanics of the pendulum.
Thank you for this catch. We have revised the MS to change 60 for 30 degrees.
I’m also wondering why the authors did not consider using the period of a physical pendulum instead of, or in addition to, a simple pendulum. I think this should be discussed because, at first glance, it is unlikely that the brachiation dynamics of human would follow a simple pendular motion anyway. Inertia is likely to play a certain role.
We completely agree with the reviewer that a physical pendulum exists as a much more accurate model for our human brachiators than does a simple pendulum. That being said, there are anthropometric measurements that we do have not collected, and cannot re-measure in order to accurately model our humans as a physical pendulum. Brachiation, from a historical perspective has always alluded to the continuous interchange of a simple pendulum. As such, we wanted to test these long-standing biological questions regarding this behavior in humans. We have added a limitations section addressing this consideration.
l.117, In the possible factors assessed, I’m wondering why you didn’t take into account the distance between the CoM and the point of contact. I guess that arm length can be considered as a proxy for this, but postural adjustments, such as flexing the legs for example, could have a direct influence on the position of the CoM. Did you abserve this kind of postural strategy?
Thank you for this comment. In this paper we sought to assess the anatomical correlates that drive energy recovery. This distance between the COM and the point of contact, or as we call it L in our MS, is a distance that can be altered behaviorally and is difficult to reconstruct from osteological metrics. Therefore, we did not include this in the possible factors assessed.
Material and Methods
l.124, The average mass of the participants should be reported, as well as the sex ratio.
Thank you for this comment. We have added these to the MS.
l.156, Stride frequency is the inverse of stride duration (not stride length).
Thank you for catching this. We have updated this in the MS.
l.160-162, To be clarified according to the definition of the stride.
We have updated this in the MS.
l.164, I think it would be better to develop further on the calculation of the COM position.
Thank you for this comment. These are well established methods are not developed by the authors of this paper. We have provided citations for reference in how COM is derived from acceleration data.
l.176, I would keep the term “plane” instead of “axis”.
Thank you for catching this. We have updated this in the MS.
l.180, Change “9.81 m/s” by “9.81 m/s2”
Thank you for catching this. We have updated this in the MS.
l.185, The multiplication sign looks like the letter “x”
Thank you for catching this. We have updated this in the MS.
l.199, Why did you keep the models with the delta AICs under three? Is it arbitrary, or based on literature?
Thank you for this comment. We followed the methods outlined in
Alex M. Rubin, Richard W. Blob, Christopher J. Mayerl; Biomechanical factors influencing successful self-righting in the pleurodire turtle Emydura subglobosa. J Exp Biol 15 July 2018; 221 (14): jeb182642. doi: https://doi.org/10.1242/jeb.182642
The larger the delta AIC the greater the deviation from the best model. Smaller delta AIC values are more conservative and better explain the variation using the variables that actually drive differences.
Results
l.203, The first sentence is odd. I think a "and" is missing.
Thank you for catching this. We have updated this in the MS.
l.206, The unit for the grip strength is missing.
Thank you for catching this. We have updated this in the MS.
l.209, The unit for the stride length is missing
Thank you for catching this. We have updated this in the MS.
l.211. Is the observed pendulum period calculated per stride, or per swing phase?
The observed pendulum period is calculated per stride.
In Table 1, I would find interesting to add the absolute grip forces.
Thank you for this comment. We have provided absolute grip forces in the supplemental data sheet. All the data presented in Table 1 are variables we statistically tested for.
In Table 1, change “Medialateral” for “Mediolateral”
Thank you for catching this. We have updated this in the MS.
In Table 1, The unit of the mediolateral and vertical excursions are in °, but it should be in meter.
Thank you for catching this. We have updated this in the MS.
Figure 1.c) and Figure 2.b) It would be informative to include the “handfall pattern” on the graphs, at least vertical lines showing the timing of the lift-off for each hand and the double support.
Thank you for this suggestion. We have added this chart to Figure 2.
Discussion
l.268, References are missing about the specific anatomical features related to brachiation in hominins.
The citations have been added.
l.276, You write “The sharp dip in kinetic energy at mid-swing”, while in the legend of Figure 1 you write “…individuals tend to reduce velocity in the middle of the stride”. So, this is really confusing because the mid-stride should be a phase of double support, while mid-swing should represent a period where there is only one support. It is important to define the stride and to refer to the stride (i.e. two swing periods) or to the step (i.e. one swing period), when appropriate.
We thank the reviewer for this catch. We have gone through and all references of mid-swing have been changed to mid-stride.
l.299, Is such a strategy called “brachiation”? It looks more like a “flexed elbow forelimb swing” as it is defined in Hunt et al. 1996 (Hunt KD, Cant J, Gebo D, Rose M, Walker S, Youlatos D (1996) Standardized descriptions of primate locomotor and postural modes. Primates 37(4):363-387). To be discussed.
While we agree that such a behavior reflects the “flexed elbow forelimb swing” described by Hunt and colleagues, we do not believe that such sub-categorization is overall helpful. Quantification of primate locomotion has artificially inflated perceived diversity of the order due to the increased sub-categorization of behaviors. Instead, we will use brachiation broadly to describe the gross motor pattern of forelimb-driven propulsion and body weight support.
l.309, What about security in addition to flexibility?
This has been added and expanded. Thank you for the suggestion.
l.327, Some references for this convergent morphology should be added here.
References have been added
I think that a figure showing the relationship between dimensionless speed and energy recovery in humans and in gibbons and/or siamang would be highly valuable and informative in this study (even if it's not significant in humans). I believe that the authors have all the relevant data to do so.
Thank you for this suggestion. We have added this chart to Figure 1.

Round 2
Reviewer 3 Report
Dear Editor, dear authors,
You provided me with the opportunity to revise a second time the manuscript entitled « How pendular is human brachiation? When form does not follow function. » by Young et al.
This revised manuscript is much improved. This new version of the manuscript will be of general and broad interest to the field of biological anthropology. However, I have still a small additional concern when the authors compare their results to the published results about gibbons. The authors now use the stride as the general reference unit throughout their manuscript, while in previous studies, the reference unit is often the swing phase. Authors should be careful to compare the same things. I'm not sure it's always the case.
One important thing to correct is also the Figure 1. Although the authors changed the legend according to my comments, the Figure 1 has not been updated accordingly (it’s the same than in the previous version, it's maybe a mistake when uploading the new version). So, this needs to be corrected.
Otherwise, I have just minor comments:
- Remove the parentheses () before the first equation in the introduction.
- Now that this equation is the same than the first one in the material and methods (expected pendular period), you can just refer to this equation without writing it again.
- The equations for EP and EK should be aligned.
- For the percent recovery equation, I would directly include the positivity of the increments in the equation, with a "+" after each increment symbol.
- In the discussion you added the following sentence: “As such, we test long-held form-function hypotheses about what makes an anatomically “ideal” brachiator capitalizing upon natural variation that exists in modern human populations” I’m not sure to understand your point here and specifically the end of the sentence. Your study includes 9 individuals, so from this sample size I don’t think you can postulate anything about the natural variation in modern human populations. I would just stop the sentence after “ideal” brachiator.
- In the discussion, 3rd paragraph, sentence starting with “Upon cursory visual inspection…” I think there is still a confusion here. I don’t have the new figure 1 in general, so I cannot really check, but be careful that in the paper of Bertram and Chang 2001, authors are considering one swing period as the reference, not a full stride. I believe it is the same in Byron et al. 2017, they consider the swing phase only (almost equivalent to the simple support phase only). To compare your values with them, you should use the swing phase only and thus half the pendulum. I’m still wondering whether the same things are compared here.
- With regard to the interpretation of the differences between humans and non-human primates about the expected pendular period, the swing phase (i.e. the simple support phase) is much shorter in humans than in gibbons and the swing phase is much shorter than the stance phase in humans (accroding to your duty factor). Hence, it looks like humans do not actually follow the mechanics of the pendular motion. They increase the control of their swinging phase by making it very short and having long stance phase (and long double stance phases). The duty factor is much closer to 50% in gibbons, making it a closer match to the natural pendular period in these specialised brachiators. Then they can rely on pendulum mechanics, humans cannot. I think this idea should be added in the discussion. Anyway, following the corrections provided after the previous version, it looks now much more logic to me that humans have shorter pendular periods than expected according to natural pendulum because our body (upper limb) is not built up to support such tensile forces.
Author Response
Dear Editor, dear authors,
You provided me with the opportunity to revise a second time the manuscript entitled « How pendular is human brachiation? When form does not follow function. » by Young et al.
This revised manuscript is much improved. This new version of the manuscript will be of general and broad interest to the field of biological anthropology. However, I have still a small additional concern when the authors compare their results to the published results about gibbons. The authors now use the stride as the general reference unit throughout their manuscript, while in previous studies, the reference unit is often the swing phase. Authors should be careful to compare the same things. I'm not sure it's always the case.
One important thing to correct is also the Figure 1. Although the authors changed the legend according to my comments, the Figure 1 has not been updated accordingly (it’s the same than in the previous version, it's maybe a mistake when uploading the new version). So, this needs to be corrected.
Corrected. Thank you for this catch.
Otherwise, I have just minor comments:
- Remove the parentheses () before the first equation in the introduction.
Corrected. Thank you for this catch.
- Now that this equation is the same than the first one in the material and methods (expected pendular period), you can just refer to this equation without writing it again.
Revised.
- The equations for EP and EK should be aligned.
Corrected. Thank you for this catch.
- For the percent recovery equation, I would directly include the positivity of the increments in the equation, with a "+" after each increment symbol.
Revised.
- In the discussion you added the following sentence: “As such, we test long-held form-function hypotheses about what makes an anatomically “ideal” brachiator capitalizing upon natural variation that exists in modern human populations” I’m not sure to understand your point here and specifically the end of the sentence. Your study includes 9 individuals, so from this sample size I don’t think you can postulate anything about the natural variation in modern human populations. I would just stop the sentence after “ideal” brachiator.
Revised.
- In the discussion, 3rd paragraph, sentence starting with “Upon cursory visual inspection…” I think there is still a confusion here. I don’t have the new figure 1 in general, so I cannot really check, but be careful that in the paper of Bertram and Chang 2001, authors are considering one swing period as the reference, not a full stride. I believe it is the same in Byron et al. 2017, they consider the swing phase only (almost equivalent to the simple support phase only). To compare your values with them, you should use the swing phase only and thus half the pendulum. I’m still wondering whether the same things are compared here.
Thank you so much! This was an important point. We can confirm that all pendulum period data is based on a stride (not just the reference limb contact). We have replaced the trace with a full stride.
- With regard to the interpretation of the differences between humans and non-human primates about the expected pendular period, the swing phase (i.e. the simple support phase) is much shorter in humans than in gibbons and the swing phase is much shorter than the stance phase in humans (accroding to your duty factor). Hence, it looks like humans do not actually follow the mechanics of the pendular motion. They increase the control of their swinging phase by making it very short and having long stance phase (and long double stance phases). The duty factor is much closer to 50% in gibbons, making it a closer match to the natural pendular period in these specialised brachiators. Then they can rely on pendulum mechanics, humans cannot. I think this idea should be added in the discussion. Anyway, following the corrections provided after the previous version, it looks now much more logic to me that humans have shorter pendular periods than expected according to natural pendulum because our body (upper limb) is not built up to support such tensile forces.
Thank you for this suggestion. While it is an interesting thought, we believe we have addressed such information already as written. Further, we did not formally run any statistics on duty factor. As such, we would prefer not to speculate.
